# Humility and Realism in Quantum Physics and Metaphysics

Damiano Bondi

Department of Economy, Society, Politics, University of Urbino, 61029 Urbino, Italy; damiano.bondi@uniurb.it

**Abstract:** The aim of this paper is to discuss some of the main philosophical and metaphysical implications of quantum physics, especially those which concern the issues of epistemic humility and ontological realism. My thesis is that the impossibility of reaching an objective knowledge of nature does not imply the renunciation of ontological realism, but rather encourages scientists to adopt an attitude of epistemic humility. The argument firstly presents the main theories of quantum physics currently discussed, focusing on the measurement problem and its ontological implications. Afterwords, the issues of objectivity and realism are properly addressed. In the end, we discuss statistics as the new form of scientific epistemology, along with the concept of potentiality as the fundamental category of quantum metaphysics. Throughout, we establish some parallelisms between quantum physics theories and theology to show that, when human beings investigate the foundations of reality, some thought patterns, some core problems, and some possible solutions resemble one another, regardless of the specific perspective and language with which they are formulated.

**Keywords:** quantum physics; metaphysics; realism; humility; potentiality

> Quantum mechanics is ontologically revolutionary,
> even if we can't say exactly what form the revolution takes.
>
> (Lewis 2016, p. 71)

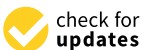



## 1. Introduction: Natura Facit Saltus

Since Planck first formulated the hypothesis of 'quantum of action' in 1900, a true revolution began in physics, which (in the words of one of its leaders) "in many respects resembles the scientific revolution of the sixteenth and seventeenth centuries" (Bohr 1960), and which is still ongoing. In the future, we will likely reach a new coherent paradigm, much like the Newtonian system was until the twentieth century. However, currently, the situation strongly resembles the battle among different cosmologies in the early modern era.

Obviously, this paradigm shift involves philosophical, epistemological, and metaphysical issues; in this article, we will discuss some of them. Indeed, we are convinced that the best way to remain faithful to Aristotelian empiricism does not mean stubbornly preserving classical metaphysics while ignoring the profound changes in physics, but striving to understand contemporary physics and then reflecting on its metaphysical implications. On the other hand, one can also say that "if an interpretation of quantum mechanics cannot yield a coherent metaphysical picture of the world, then it cannot be regarded as an adequate descriptive theory" (Lewis 2016, p. 82); that is, metaphysical coherence can be seen as an epistemological criterion, playing a role in the 'battle' of physics.

First, we will try to summarize the main general ontological options suggested by quantum physics; secondly, we will explore some specific categories and problems in the ongoing debate about quantum metaphysics, focusing on the issues of humility and realism; throughout, we will establish some parallelisms between quantum physics theories and theology, simply to show that, when human beings investigate the foundations of reality, some thought patterns, core problems, and possible solutions resemble one another, beyond the specific perspectives and languages with which they are formulated.

## 2. How Many Possible Quantum Worlds Do We Have?

According to what John Bell said in 1986, we should consider "Six Possible Worlds of Quantum Mechanics" (Bell 2004, p. 181), corresponding each to a different interpretation of quantum phenomena. In combining his picture with the more up-to-date picture sketched by P. J. Lewis, the current situation could be summarized through presenting four main options, plus some radical proposals.

(a) *Copenaghen Interpretation*. The standard interpretation of quantum phenomena was developed during the 1920s by Niels Bohr, Werner Heisenberg, Max Born, and others, and was culminated in 1927 in the Solvay Conference in Brussels. Briefly, the Schrödinger equation of the wave function is interpreted as describing (if squared) the *probability* of the existence of a particle in a determinate space region; but, if we experimentally measure where the particle is, we will find it in a precise point, as if the wave function would 'collapse' in a point. More generally, since the Schrödinger equation is a linear differential equation, any linear combination of its solutions is still *a* solution of the Schrödinger equation. Therefore, a wave function can be a linear combination of wave functions representing different states (*superposition principle*). This leads to the *measurement problem*, which asks how can *precise* particle properties (such as position, velocity, or energy) emerge when a measurement is made? According to the Copenhagen Interpretation, the *measurement itself* is the cause of the wave function collapse, but how it can do so remains a mystery (Schlosshauer et al. 2013). It seems that "physical systems evolve following the Schrödinger equation, *except* during a measurement" (Lewis 2016, p. 49); is measurement an event that is *ontologically* different from any other physical event? To try to solve this problem, other theories have more recently been proposed.

(b) *Spontaneous Collapse Theory* (or GRW Theory, acronym from acronym from Ghirardi et al. (1986), who developed it). Particle phenomena can be described as wave functions, which evolve following the Schrödinger equation *and* another probabilistic law. This law describes the (very rare) possibility of a spontaneous and random 'hit' in which the particle state suddenly becomes more localized. Note that this eventual 'hit' is independent from the measurement. The measurement 'simply' greatly increases the probability that collapse will occur, since the measuring instrument (which is correlated with the observed particle) is macroscopic—and a macroscopic object is itself composed of countless correlated particles, such that the spontaneous localization of just one of them causes the localization of the entire system. The initial state of the system here determines only the probability distribution for the point on which the hit could be centered, but neither the exact point nor the actual occurrence of the wave function collapse.

(c) *Hidden Variable Theory* (or Bohm's Theory (Bohm 1952), since David Bohm first presented it in 1952). If GRW ontology is wave-like, Bohm's ontology is clearly dualistic. The world is constituted by particles *and* wave-shaped fields 'pushing' those particles around. The 'hidden variable' is the position of the particles, which can be revealed only after the measurement. What we can know *before* the measurement, i.e., independently from it, is the probability of finding the particles in a specific space region. Were the position of the particles not 'hidden', however, the whole system would then be completely known, since its evolution is entirely deterministic.

(d) *Many-Worlds Theory*: first proposed by Everett (1957) and divulgated by Graham DeWitt, according to this theory, the world is constituted by infinite separate branches of reality. The wave function of every measured quantum physical process includes the measurement itself, therefore every possible outcome must be interpreted as a 'branch' in which the world is actually split after the measurement. Note that even the eventual observer is split in multiple versions of 'themselves'. In each branch, particles mutually interact in a classical physical way, but there is no interaction among the branches. Therefore, each branch is like an independent *world*. In this theory, *every*

possible outcome of a measurement actually occurs (producing its own branch of reality), and there are no real differences of probability among the single outcomes.

In addition of these four main 'possible Quantum Worlds' (even if the last one postulates many), there are some alternative theories, or radical interpretations. We will mention only those which, according to our opinion, present the most important metaphysical implications.

(a)　*Bare Theory*. Developed by David Z. Albert and Jeffrey Barrett, this theory radicalizes the Many-Worlds Theory, postulating *one world*, but one that is completely indeterminate and unknowable. Indeterminacy, which the collapse postulate seems to eliminate, is actually at the core of the entire universe; quantum phenomena indicate it as the undetectable *noumenon* beneath the phenomenal determinate surface. The Bare Theory can be seen as the 'physically updated' version of various skeptical positions in the history of philosophy (as well as various Matrix-style cinematic dystopias), but it is perhaps Barrett himself who has established the most effective parallelism. He writes that Bare Theory's conception of experience "makes Descartes's demon and other brain-in-a-vat stories look like wildly optimistic appraisals of our epistemic situation" (Barrett 1999, p. 94).

(b)　*Quantum Consciousness Theories.* The obvious objection that can be made to the Bare Theory asks then why do we experience determinate objects (starting from ourselves)? One of the possible answers is that there are conscious metaphysical entities (demons, gods, or morally neutral ones) which cause the wave function to collapse. This seems indeed the solution proposed by Federico Faggin, Giacomo M. D'Ariano, Donald D. Hoffman, and other authors, who we have grouped together under the label of 'Quantum Consciousness Theories' (see for instance D'Ariano and Faggin 2022). According to these theories, instead of trying to explain how consciousness can arise from life, and life from matter, we should reverse the cause and effects, and hypothesize that *consciousness comes first*, and that life and matter arise from it. This explanation, which seems inadmissible only because we are culturally conditioned by Western materialist reductionism, would actually be the simplest to account for quantum phenomena (such as entanglement and interference) and, at the same time, for the apparent exceptional nature of living systems. Faggin's theory, in particular, is to all intents and purposes a metaphysical system—and perhaps it is no coincidence that his father, Giuseppe Faggin, was one of the most important international scholars of Plotinus' thought. Following the line of panpsychism (integrated with the Quantum Information Theory developed by D'Ariano), Federico Faggin postulates the existence of conscious quantum entities, the *Seities*, which emanate hierarchically from One (the conscience of all that exists) and which communicate with each other; this 'semantic exchange' would create reality as we know it (see Faggin 2022). The parallel between seities and angels is almost obvious, but perhaps less obvious is another theological suggestion, namely the parallelism with the role of the different persons in the Trinity. The doctrine of the Trinity fits well with the Quantum Consciousness Theories only if conceived in a neo-platonic way (see Moreschini 2021). Nevertheless, we can establish a wider parallelism between Trinitarian Ontology and Quantum Metaphysics in general. In fact, while reflecting on Trinitarian theology, Joseph Ratzinger wrote in 1968 that an "approach of modern day physics may offer us more help here than Aristotelian philosophy", since in quantum physics there is "the idea of a being that has no substance but is purely actual, whose apparent "substantiability" really results from the pattern of movement of superimposed waves". According to Ratzinger, Schrödinger's theory in particular "remains an exciting simile for the *actualitas divina*, for the fact that God is absolutely "in act", and for the idea that the densest being—God—can subsist only in a multitude of relations, which are not substances but simply "waves", and therein form a perfect unity and also the fullness of being" (Ratzinger 2004, pp. 174–75).

(c) *Retrocausal Theories and Flashy Theories.* Starting from the possibility to argue for backward causation in microphysics via the temporal symmetry (Price 1996), and drawing the most extreme consequences from John A. Wheeler's (1978) delayed-choice experiments, some physicists have theorized the possibility that the behavior of the particle in the present is causally influenced by the *future* measurement. Most of these theories are built on a Bohmian model (Aharonov and Vaidman 1990; Sutherland 2008; Wharton 2010), in which the 'hidden variable' is not really 'hidden', but only 'not yet caused by future events'. There are also potentially retrocausal elements in some versions of the GRW theory called 'flashy' theories, like the one developed by Roderich Tumulka (2006). In this line, we should consider that each "piece of matter is a galaxy of [wave function collapse] events" (Bell 2004, p. 205); due to the fact that the center of a quantum 'hit' has a precise space–time location, we can conceive the macrophysical world as being made up of countless spontaneous microphysical collapses occurring at any moment. The 'glitchy' picture of the world this theory enables is not so far from the one suggested by the most radical interpretations of the many-world theory (such as those by David Z. Albert and David Wallace: see Albert and Lower 1996; Wallace 2010), according to which "macroscopic objects undergo branching events all the time, many times per second, based on quantum interactions with their environment" (Lewis 2016, p. 139). But then, how can the objects of at least one world remain in existence over time, especially if a necessary causal connection between two phenomena is not guaranteed? This is exactly the same question that inspired one of the major currents of theology and philosophy in the Middle Ages and early modern ages, namely *occasionalism*, according to which, in order to justify the world as it is, there must be a sort of 'continuous creation' of the world by God. It is certainly no coincidence that some scientists have recently identified a similarity between issues involved in what we have called 'flashy' theories and al-Ghazālī's theological occasionalism, resulting in discussions about "quantum occasionalism" (Harding 1993; Harman 2016; Weir 2020).

(d) *Relational Monism.* Also known as 'ontic structural realism' (OSR), this theory has been developed mainly by James Ladyman (French and Ladyman 2003), Don Ross, and Steven French, and endorsed also by Carlo Rovelli (1996) and N. David Mermin (1998). Consciously rejecting Leibniz's doctrine of the identity of the indiscernibles (Ladyman and Ross 2007), the basic metaphysical principle of the universe, here, is a *structure of relations*, which is a priori with regard to any individual entity, and which best accounts for the emergent properties of particles, such as entangled spins. These properties, in fact, would not be properly 'emergent', rather the opposite; it is the individual particles that emerge from the underlying all-ruling Relation. If we add consciousness and will to this Relation, we can establish the same theological parallelism as we have for the Quantum Consciousness Theory (see above, letter f). More generally, this way of thinking may recall the *Gestalt Theory* (see Amann 1993), or even the environmental ontology implied in deep ecology, as metaphysically conceived by Arne Naess (see Naess 1976; Oppermann 2003). The Relational Monism has a serious coherence problem, namely the possibility to speak about relation(s) without entities which relate to each other. We can certainly say that quantum properties such as spin entanglement are emergent ones, since they are irreducible to their parts (in that case, irreducible to the properties of individual electrons); in this sense, we can certainly speak about quantum holism—entanglement can indeed be seen as an almost archetypal model of an 'emergent property'. But holism does not entail relational monism ipso facto. There can be irreducible properties emerging from the relation between ontologically separate entities. Furthermore, Relational Monism does not fully solve the measurement problem—which is at the root of every alternative quantum theory to the Copenhagen interpretation.

### 3. Realism without Objectivity, or the Epistemic Humility in Quantum Physics

At this point, we delve deeper into the following question: do the main alternative interpretations *really* solve the measurement problem? Well, at a closer look, the measurement postulate seems to still be there, even if reshaped. In the Spontaneous Collapse Theory, measurement ends up being the *only* physical event which determines a non-spontaneous collapse. In the Hidden Variable Theory, it is the act of measurement which *reveals* where the particles *really* are—and in its retrocausal version, measurement is so important that it can influence events in the past. In the Many-Worlds Theory, without measurement, there would not even be a ontological division. In any case, measurement remains a physical event characterized by a certain degree of *exceptionality* when compared to others. This exceptionality is probably due to the fact that measurement is a physical event with an inherent psychic element; there is intentionality in measuring something, and/or awareness in doing so. Without this element, we could not define an act as a 'measurement'. Even if we focus only on the instrument and talk about 'data recording', this recording has always been prepared by scientists for scientific cognitive purposes, and the result can be considered as 'not decided' until it is read by a conscience. This feature is *minimum* human, meaning that consciousness is recognized by all as a human characteristic at the very least. That is why, when Quantum Consciousness Theories suggest attributing it to the entire universe—transforming it into a metaphysical category—they are subjected to the criticism of practicing nothing different than anthropomorphism. Relational Monism, with a similar move, postulates an all-encompassing principle of correlations behind and beyond the 'measurer–measured' correlation. The Bare Theory recognizes the centrality of the conscious/mental element precisely in declaring it as a total illusion.

In summary, as Fritz London and Edmond Bauer pointed out in 1939, "at first sight it would appear that in quantum mechanics the concept of scientific objectivity has been strongly shaken [...] it looks as if the result of a measurement is intimately linked to the consciousness of the person making it, and as if quantum mechanics thus drives us toward complete solipsism" (London and Bauer 1983, p. 258). However, they themselves recognized that this did not happen; the scientific community continues to communicate, cooperate, and share theories, hypotheses, and results. How is it possible?

We cannot, nor do we want to, discuss here the fascinating phenomenological solution first proposed by London and Bauer and rediscovered today by Steven French—according to which, briefly, the self-awareness of the observer enters the quantum superposition and separates the object from the subject (see French 2002). Rather, we identify two main general elements that allow us to overcome the crisis of scientific objectivity, or rather the crisis of the myth of the objective knowledge of nature. These elements are realism and statistics. We will discuss the latter in the next paragraph, and the former here.

For sure, we can agree with London and Bauer in recognizing that the possibility to reach a complete *objective* scientific knowledge of the world is declared (by all current interpretations of quantum phenomena) as nothing but a myth of modern science. This myth was based on the presumptuous illusion of being able to view the world 'from God's point of view', as if the observer could remove themselves from the equation of the world, and could thus attain knowledge of the thing-in-itself. As Werner Heisenberg notes, "it may be said that classical physics is just that idealization in which we can speak about parts of the world without any reference to ourselves. Its success has led to the general idea of an objective description of the world" (Heisenberg [1958] 1962, p. 55). We can have the illusion of objectivity only if we stay at the level of mesophysical phenomena (and not even all of them), where the implications of quantum mechanics can be considered irrelevant, but, as soon as we start to explore the microphysical world, everything changes. The phenomena no longer 'respond' to expectations; they do not behave as they were expected to behave. We can, at most, make probabilistic predictions about them, and this indeterminacy is not due to the imprecision of the experiment or the instruments (as are the errors in classical physics); rather, it is inherent to the observed reality, and therefore ineliminable. Following the work of James van Cleve and David Matthews, we can define the awareness of the

impossibility of reaching scientific objectivity as "epistemic humility" (Matthews 2006; Van Cleve 2011).

What implications does epistemic humility have on realism? First, we need to distinguish between ontological realism and (what we can call) epistemic realism. By 'epistemic realism' (which is *not* epistemological realism, since it is a form of fallibilism), we mean an empiricist attitude by which a scientist is always open to modifying their theories if the facts contradict them. Epistemic realism should be accepted and shared as an epistemic virtue by the entire scientific community, even though—or better, precisely because—we know that sometimes scientists, who are human beings, remain attached to their theories to the point of consciously forcing the facts and data to fit them. We might say that the unexpected quantum phenomena forcefully 'asked' to be interpreted by the physicists of the early twentieth century with a posture of epistemic realism. Over time, this has led the scientific community to gain another virtue, epistemic humility, thus freeing itself from the proud idealism inherited from previous centuries. As Lewis points out, indeed, "the metaphysically problematic nature of quantum mechanics is not just a matter of interpreting an obscure theory, but is a problem in the empirical phenomena themselves—in the behavior of objects in the world. The world does not conform to our classically trained intuitions" (Lewis 2016, p. 9). Heisenberg gives a good idea of the state of mind of the first theorists of quantum mechanics as follows: "I repeated to myself again and again the question: Can nature possibly be as absurd as it seemed to us in these atomic experiments?" (Heisenberg [1958] 1962, p. 42). Interference and entanglement appear as mysterious real phenomena, so much that physicists such as Heisenberg and Schrödinger, with epistemic realism, elaborate their mathematical hermeneutics as follows: the matrix mechanics, which is better suited to represent the distribution of discrete quantities (such as spins), and the wave mechanics, which instead can be seen as a method of quantization of distributed properties (such as the interference). In the same line, Einstein's argument for incompleteness (also known as the EPR argument, from the initials of Einstein, Podolsky, and Rosen), even if contradicted by Bell's theorem, is an example of epistemic humility. It establishes that quantum mechanics cannot be considered a complete description of physical reality (Einstein et al. 1935). To what extent, then, does it describe reality? In other words, what about ontological realism?

Abraham Pais tells the following famous anecdote regarding Einstein himself: "We often discussed his notions on objective reality. I recall that during one walk Einstein suddenly stopped, turned to me and asked whether I really believed that the moon exists only when I look at it" (Pais 1979). Einstein's polemical objective was a skeptical radicalization of the Copenhagen interpretation, which, however, can lead to idealistic outcomes; reality in itself cannot be known or described because it does not even exist without an observer. At first glance, it may seem that Copenhagen's interpretation of quantum phenomena not only retires the concept of objectivity, but also that of ontological realism. This is one of the reasons why alternative solutions to the Copenhagen interpretation have been proposed: to save realism, avoiding the annoying intrusion of subjective elements into scientific knowledge (represented by the measurement postulate). But is it really like that? Does the Copenhagen interpretation declare that ontological realism is nothing but a myth? And more deeply, if scientific objectivity falls, does ontological realism follow suit? This is a truly philosophical question, and we believe it can be answered philosophically, beyond the purely physical debate among different theories.

We believe also that the answer is no, the death of scientific objectivity does not imply the end of ontological realism. Not being able to fully know something 'in itself' does not in any way imply that nothing exists independently of our knowledge or perception. To stick with Einstein's example, the fact that there is something which manifests itself to our knowledge and senses like 'the moon' is not banished by the fact that what I call 'moon' does not correspond exactly to that thing. Without the idea of nature—that is, the idea that something exists independently of our will—science itself would be impossible. Ultimately, the surprise of the first quantum physicists at the strangeness of the results of the experiments cannot be explained as anything other than wonder at something that

exists independently of me and my expectations and knowledge. If something unexpected arrives, it is because it exists independently of the person who expects something else. As Heisenberg ([1958] 1962) summarizes, "Natural science does not simply describe and explain nature; it is part of the interplay between nature and ourselves; it describes nature as exposed to our method of questioning" (p. 81).

Ontological and epistemic realism can help science not to fall into objectivist reductionism on the one hand, and presumptuous idealism on the other. These two errors can be united if theologically interpreted as two ways of 'becoming like God', who creates the world from nothing (else but from oneself) and observes it from the outside.

On the contrary, realism, as we have described it, produces a form of epistemic humility which does not lead to absolute skepticism, but rather encourages scientists to investigate what nature reveals to our questioning knowledge. In other words, it encourages scientists to investigate how nature *probably* is.

## 4. Probability, Statistics, and Potentia

Schrödinger peremptorily declares that "the laws of physics and chemistry are statistical throughout" (Schrödinger 1944, p. 2). The laws of classical physics seem exact only because they describe events at the mesoscopic and macroscopic levels, which involve entities composed of an extremely large number of atoms. But that exactness is actually only a very, very high probability. 'For all practical purposes', we can say that that law is exact; but for all theoretical purposes, we must recognize that that apparent exactness is only a matter of size. This is a key point of Schrödinger's theory, in that the laws of classical physics are 'proportionate' to our sense organs. Just as these organs (including the brain) cannot be too small, under penalty of being influenced in their ordered functioning by single atoms or molecules, similarly the mesophysical laws do not take into account the behavior of the single atom, but the average behavior of many atoms, namely the general tendency. In this sense, according to Schrödinger, living beings are 'mysterious' and difficult to explain, since genes exhibit very regular behavior considering their very small size—but we cannot go into further detail here. What interests us is the concept of statistics connected to that of tendency, because it recalls some categories of Aristotelian metaphysics. Indeed, this is a common feature of all the quantum physics theories that we have seen: "quantum mechanics makes its prediction in the form of probabilities, and the success of quantum mechanics lies in the agreement between these probabilities and the relative frequencies of the outcomes we observe" (Lewis 2016, p. 130). In other words, a prediction in quantum physics is more accurate the more it identifies a tendency, a greater frequency of a specific outcome compared to others. This is a real novelty compared to Newtonian physics, and Heisenberg traces its origin in the theory of 'probability wave' formulated by Bohr, Kramers, and Slater in 1924 as follows: "the probability wave of Bohr, Kramer and Slater—he writes—meant a tendency towards something. It was a quantitative version of the old concept of *potentia* in Aristotelian philosophy. It introduced something standing in the middle between the idea of an event and the actual event, a strange kind of physical reality just in the middle between possibility and reality" (Heisenberg [1958] 1962, p. 41).

There are two ways in which the classical metaphysical concepts of potentia, actus, matter, and form can be useful to understand quantum phenomena.

1.  Since experiments have shown that all the elementary particles, with a sufficiently high energy, can be transmuted into other particles, we can conclude that the 'unity of the matter' has been proved, meaning that (as Heisenberg points out) "all the elementary particles are made of the same substance, which we may call energy or universal matter; they are just different forms in which matter can appear. If we compare this situation with the Aristotelian concepts of matter and form, we can say that the matter of Aristotle, which is mere "potentia," should be compared to our concept of energy, which gets into "actuality" by means of the form, when the elementary particle is created" (Heisenberg [1958] 1962, p. 160). More precisely,

what can be compared to a pure *potentia*, without form, is what Aristotle called hypokeimene physis, the 'nature-substratum', which can only be known by analogy, and which, according to the commentary provided by Thomas Aquinas, is "the subject of all forms" (see Aristotle, *Physics,* I, part VII, 9 and Th. Aquinas, *In I Physic.*, lc. 13, n. 9). Nevertheless, Heisenberg continues, "modern physics is of course not satisfied with only qualitative description of the fundamental structure of matter; it must try on the basis of careful experimental investigations to get a mathematical formulation of those natural laws that determine the "forms" of matter, the elementary particles and their forces". However, the experimental investigations of twentieth-century physics and the related mathematical formulations (including the wave function, or Einstein's theory of relativity) have shown that a clear distinction between matter and force, or between mass and energy, cannot be made. "Each elementary particle not only is producing some forces and is acted upon by forces, but it is at the same time representing a certain field of force. The quantum-theoretical dualism of waves and particles makes the same entity appear both as matter and as force" (Heisenberg [1958] 1962, p. 160). It therefore seems that the ancient theory of Heraclitus, according to which the unifying principle of Being is precisely a dynamic principle of Change, is once again relevant. Not by chance, Heisenberg himself names the philosopher of Ephesus in the following: "In the philosophy of Heraclitus the concept of Becoming occupies the foremost place. [. . .] But the change in itself is not a material cause and therefore is represented in the philosophy of Heraclitus by the fire as the basic element, which is both matter and a moving force. We may remark at this point that modem physics is in some way extremely near to the doctrines of Heraclitus. If we replace the word "fire" by the word "energy" we can almost repeat his statements word for word from our modern point of view". (Heisenberg [1958] 1962, pp. 62–63). The metaphysical principle of the 'tension of opposites' seems to reappear in quantum physics, not only for wave–particle and mass–energy pairs, but also for the spin entanglement, or for Heisenberg's Uncertainty Principle. Niels Bohr gave it a specific name and made it a cornerstone of the Copenhagen Interpretation: the principle of complementarity.

2.  According to Bohr, the principle of complementarity had a broader value and application than just quantum physics. It can be said that, for him, it was a true philosophical, metaphysical principle. Bohr himself designed his family coat of arms with the symbol of the Tao and the Latin motto *Contraria sunt Complementa*, and, in his writings, he often attempts to apply complementarity to biological, psychic, and cultural phenomena (see Bohr 1932, 1938). We know that Bohr was a passionate reader and admirer of his compatriot Kierkegaard, and it is not difficult to identify similarities between Bohr's theory and some Kierkegaardian themes, such as the enten-eller or the conception of "stages" (see Giannetto 2019). For sure, if Bohr had also been a theological scholar, it would have been easier to establish a parallelism between the principle of complementarity and the doctrine of the double nature of Christ, whose divinity and humanity are co-present 'without confusion or separation' (as we can read in the Chalcedonian dogmatic formula). Remaining on the level of physics, Bohr declares that "far from being inconsistent, the aspects of quantum phenomena revealed by experience obtained under such mutually exclusive conditions must thus be considered as complementary to each other in quite a novel way", for example, "any imaginable procedure aiming at the coordination in space and time of the electrons in an atom will unavoidably involve an essentially uncontrollable exchange of momentum and energy between the atom and the measuring agencies, entirely annihilating the remarkable regularities of atomic stability for which the quantum of action is responsible. Conversely, any investigation of such regularities, the very account of which implies the conservation laws of energy and momentum, will on principle impose a renunciation as regards the space-time coordination of the individual electrons in the atom" (Bohr 1938). In the same way, "the spatial continuity of light

propagation, on one hand, and the atomicity of the light effects, on the other hand, must be considered as complementary aspects of one reality. [. . .] This very situation forces us to renounce a complete causal description of the phenomena of light and to be content with probability calculations, based on the fact that the electromagnetic description of energy transfer by light remains valid in a statistical sense" (Bohr 1932). On this basis, Carl von Weizsäcker built his logic of the 'degrees of truth' using complex numbers in order to establish the probability of a physical system to have determinate alternative properties. Let us take, for example, an atom which moves in a closed box, divided by a wall into two equal parts with a very small hole in the middle (see Heisenberg [1958] 1962, p. 182). According to classical logic, the atom can be either in the right half or in the left half. Tertium non datur. But the results of the experiments (e.g., on the distribution of intensity of the light which has been scattered by the atom) show that this is not the case, as there are other possibilities, which are mixtures of those two, due to the interference of probabilities and to the complementarity wave–particle of light phenomena. Weizsäcker proposed the use of a complex number to measure the 'degree of truth' for any simple statement in an alternative, like 'the atom is in the left half of the box'. "Each pair of complex numbers referring to the two parts of the alternative represents, according to Weizsacker's definitions, a "statement" which is certainly true if the numbers have just these values; the two numbers, for instance, are sufficient for determining the intensity distribution of scattered light in our experiment" (Heisenberg [1958] 1962, p. 183). We can have, here, a logical account for the principle of complementarity: if the proposition "the atom is in the left half" is true (or false), then the proposition "it is true that the atom is in the left half" is also true (or false), but if the proposition "it is true that the atom is in the left half" is false, then the proposition "the atom is in the left half" is *not* false, but 'not decided'. Note that ""not decided" is by no means equivalent to the term "not known". "Not known" would mean that the atom is "really" left or right, only we do not know where it is. But "not decided" indicates a different situation, expressible only by a complementary statement" (Heisenberg [1958] 1962, p. 183). If we want to 'decide' if it is true or false that the atom is in the left half, then the complementarity 'collapses' in one of the alternatives, but this does not mean that the other alternative becomes false. We are simply on another logical level. What about the *ontological* level, then? In other words, to which (ontological) state does the (logical) statement of complementarity refer? It is precisely here that some concepts of Aristotelian metaphysics can be useful again. Since complementary statements describe coexistent situations or "coexistent states", in which "every state contains to some extent also the other "coexistent states"" (Heisenberg [1958] 1962, p. 185), then it is preferable to replace the concept of 'state' with that of potentia; "the concept of "coexistent potentialities" is quite plausible, since one potentiality may involve or overlap other potentialities". Each quantum superposition state of the wave function can be viewed as a coexistent potentiality. Thus, understood, the concept of 'coexistent potentialities' would then form (in Heisenberg's words) "a first definition concerning the ontology of quantum theory" (Heisenberg [1958] 1962, p. 185). This conclusion is not so far from that of John Bell, who introduced the term "beables" in order to describe those properties of a physical system which pre-exist measurement, and which therefore cannot be observed as such; they are therefore not observable, but "beables" (Bell 2004, p. 52).

## 5. Conclusions

The tremendous technological progress of recent centuries has allowed us to explore the microscopic and macroscopic worlds, meaning the worlds which are not at our same dimensional scale; in both, instead of reaching the full knowledge of Being, we have encountered limits. We thus discovered that what we believed to be an objective knowledge of nature was actually a statistical approximation, proportionate to our physical size.

In Bohr's words, "we may say that the suitableness of the causal space-time mode of description for the ordering of our usual experiences depends only upon the smallness of the quantum of action relative to the actions with which we are concerned in ordinary phenomena. Planck's discovery has brought before us a situation similar to the one brought about by the discovery of the finite velocity of light. Indeed, the suitability of the sharp distinction between space and time, demanded by our senses, depends entirely upon the smallness of the velocities with which we have to do in daily life compared with the velocity of light" (Bohr 1929). The awareness of the limitations of our condition and our gnoseological possibilities, on the one hand, can help scientists achieve the epistemic virtue of humility; on the other hand, it does not entail either abandoning ontological realism or losing confidence in the positive value of scientific knowledge. We have seen how reflections on surprising quantum phenomena are interesting not only for physics, but also for metaphysics; additionally, certain categories of metaphysics and even theology can be found, adequately reformulated, in physical investigations as well. This is because human beings, precisely by experiencing the limit, recognize themselves as eager to know the Whole, as an inexhaustible mystery that calls.

**Funding:** This research received no external funding.

**Data Availability Statement:** No new data were created or analyzed in this study. Data sharing is not applicable to this article.

**Conflicts of Interest:** The author declares no conflict of interest.

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
