# Peer review of "Humility and Realism in Quantum Physics and Metaphysics"

_religions, doi:10.3390/rel15060670_

Round 1

Reviewer 1 Report

Comments and Suggestions for Authors

The Autor asserts:

"Throughout, we will establish some parallelisms between quantum physics theories and theology, to show that, when human beings investigate the foundations of reality, some thoughts patterns, some core problems and some possible solutions resemble one another, regardless of the specific perspective and language with which they are formulated".

I believe that the author should specify and clarify better the meaning he ascribes to the theoretically very strong and important term "regardless" in order to understand the proposal he makes in the article

Author Response

Thanks for you kind suggestion. You are totally right. "Regardless" can be interpreted in a negative way, as I would consider dismissible and useless the specific methodologies used by different disciplines. I replaced that word with a more neutral and positive term: "beyond".

Reviewer 2 Report

Comments and Suggestions for Authors

The paper is interesting and has some good ideas. Unfortunately, some of the claims are much too bold and not sufficiently backed up by arguments. Not all the relations between concepts and lines of thought are sufficiently explained. The paper is also not sufficiently engaging with debates on scientific (anti) realism. These problems may be fixable but would require a very serious revision that would change the paper profoundly.

Some examples of problems in the paper.

- 128: Your claim that quantum consciousness theories are impeded by Western materialist reductionism is too quick. Firstly it is not at all clear what western materialist reductionism means here. Second, various views of Christianity that excluded angels or demons and merely posit God and the physical universe exclude this as well.

- 141: The author should be careful here to see parallells between quantum consciousness theory and Trinitarian thought. On classical understandings of the Trinity, the three persons are not emanations that came forth at some point but exist eternally in the godhead. It's not clear how well this matches with quantum consciousness theories as these seem te presuppose a hierarchy in spiritual beings.

- 145: It is not clear how the quote from Joseph Ratzinger fits with the quantum consciousness view.

- 165: The parallells with occasionalism are again too quick. It is not clear if occasionalists allow for future causal impact on the present. If so, this should be elaborated on by the author.

- 189: relation to deep ecology is again not sufficiently clear.

- 219: It is not clear whether this really is anthropomorphism. This rests on the assumption that the model for consciousness is human. Defenders of the quantum consciousness theory would likely deny this.

- 242 vv. The move towards rejection of scientific objectivity is again too quick. There is ample literature on how scientific realism is possible with fallible human capacities which the author overlooks. It is also not clear from the discussion that the measurement probably cannot be solved in the future.

-258: the 'epistemic realism' seems more like a form of fallibilism.

Comments on the Quality of English Language

English seems fine. Minor proofing required.

Author Response

Dear reviewer,

thanks for your comments and suggestions. I accepted most of them, refused others. Here the details and the motivations:

  • 128: "Your claim that quantum consciousness theories are impeded by Western materialist reductionism is too quick". I think this is a misunderstanding: it is not my claim at all, it is the QuantumConsciousnessTheory defenders' claim. I do not endorse that Theory, I simply present and discuss it philosophically. I putted "According to these authors" at the beginning of the sentences in order to be clear.
  • 141 and 145: "The author should be careful here to see parallells between quantum consciousness theory and Trinitarian thought. On classical understandings of the Trinity...". You are right, the parallelism is too quickly and superficially exposed. I have clarified that it works only if referred to a neoplatonic understanding of the doctrine of Trinity, while the wider parallelism (following Ratzinger) regards Quantum Methapysics in general.
  • 165: the reference to occasionalism only regard what we have called flashy theories, not the retro causal ones; I have better specified this point.
  • 189: within the deep ecology there is a clear relational ontology (Naess calls it "Ecosophy T"), but unfortunately there is no space in the article to fully discuss that. I added two bibliographical references if someone wants to.
  • 219: "It is not clear whether this really is anthropomorphism. This rests on the assumption that the model for consciousness is human. Defenders of the quantum consciousness theory would likely deny this". This is a very crucial critics, thanks for having underlined this problem. A proper article would be necessary to proper address it, in any case I added some sentences indicating the way to potentially solve it.
  • 242: regarding the possible future resolution of the measurement problem, you are partially right. I only underlined at the very beginning that "Probably in the future we will reach a new coherent paradigm", without indicating how or what. I have no fully competences in that. Regarding the problem of realism and objectivity, my main thesis here is exactly to save realism without scientific objectivity (see the title of chapter 3), i.e. to save realism with a form of fallibilism.  
  • 258: yes, it is a kind of fallibilism. I added a sentence to explicitly declare it.

Reviewer 3 Report

Comments and Suggestions for Authors

My main problem with this manuscript is that there is so little on religion for a “science and religion” paper. The short references to anything religious seem to have been added just for appearance and should be worked out more fully rather than simply mentioned before moving on. So too with epistemic humility. There are four references to “mystery” in the manuscript and those too could worked out more since the manuscript means on that point.

Here are a few more specific points:

Page 1/ line 30: The reference to Kuhn is dated and the point should be reworked without reference to “paradigm shifts” or more fully explained.

Page 2/ lines 54-70: it should be brought out that the wave function is a mathematical formalism and that particles under this view are not fuzzy waves.

Page 4/ line 172: The authors might mention that occasionalism was also present in Medieval Christianity.

Page 6/ first paragraph: empiricists accept that there is a world independent of our experience — they assert that all we know of it is what we experience.

Page 6 / line 267: “idealism” might not be the best word here since you are talking about “realism” in this paragraph. Maybe just “ideal” conveys what the authors mean.

Page 7/ lines 314-317: I’m not sure if the authors are dealing with theology here, but if they are they should work it out more.

Page 8/ line 401: any parallel between Bohrian complementarity and the two natures of Christ should be worked out more.

References: There are references to P. J. Lewis and Matthews but no works are listed in the Reference section. Heisenberg 1962 is not in References. There may be other cases of this, but those were the ones I noticed. 

Here are a few typos: 

p. 2 / line 49: R.

        p. 4 / line 193: add "to" between "relate" and "each"

p. 5/ line 209: comma after “even”

Author Response

Thanks for you comments.

Generally speaking, I tried to write an article having in mind, as potential reader, a scientist which is skeptical about religion and theology. One of my aims here is to show how Quantum Physics can be open to establish mutual links with theology and religious sciences. Much more than Newtonian mechanics.

That is why I tried to be robustly rooted in my physical references, suggesting then some theological parallelisms. It is a conscious rhetorical choice.

If I had contributed to lay the foundations of an interdisciplinary dialogue, I would have succeeded in my intent.

Regarding you points: I removed Kuhn, which is effectively dated; I specified the Medieval time in the "occasionalism" suggestion.

Thanks for having detected the errors in bibliography, I fixed it.

Round 2

Reviewer 2 Report

Comments and Suggestions for Authors

The paper improved. I still believe that the paper is too bold and some of the claims are not sufficiently backed up. This may be work for a follow-up paper.

Comments on the Quality of English Language

Proofreading required

Author Response

Thanks for your precious comments. I hope now the paper has also been improved from a linguistic point of view: as suggested, language has been extensively and professionally revised.